# Management of Invasive Infections in Diabetes Mellitus: A Comprehensive Review

Anjum Khanam [1,2,†], Gavirangappa Hithamani [1,3,†], Jayapala Naveen [1,†], Seetur R. Pradeep [1,†], Susmita Barman [1,*] and Krishnapura Srinivasan [1]

1 Department of Biochemistry, CSIR—Central Food Technological Research Institute, Mysore 570020, Karnataka, India
2 Department of Clinical Nutrition, College of Applied Medical Sciences, Jazan University, Jazan 45142, Saudi Arabia
3 Department of Fruit and Vegetable Technology, CSIR—Central Food Technological Research Institute, Mysore 570020, Karnataka, India
* Correspondence: susmita.barman1987@gmail.com
† These authors contributed equally to this work.

**Abstract:** Patients with diabetes often have more invasive infections, which may lead to an increase in morbidity. The hyperglycaemic environment promotes immune dysfunction (such as the deterioration of neutrophil activity, antioxidant system suppression, and compromised innate immunity), micro- and microangiopathies, and neuropathy. A greater number of medical interventions leads to a higher frequency of infections in diabetic patients. Diabetic individuals are susceptible to certain conditions, such as rhino-cerebral mucormycosis or aspergillosis infection. Infections may either be the primary symptom of diabetes mellitus or act as triggers in the intrinsic effects of the disease, such as diabetic ketoacidosis and hypoglycaemia, in addition to increasing morbidity. A thorough diagnosis of the severity and origin of the infection is necessary for effective treatment, which often entails surgery and extensive antibiotic use. Examining the significant issue of infection in individuals with diabetes is crucial. Comprehensive research should examine why infections are more common amongst diabetics and what the preventive treatment strategies could be.

**Keywords:** diabetes mellitus; altered immune response; invasive infection; treatment strategy

## 1. Introduction

Diabetes mellitus (DM), a non-communicable metabolic condition, has been linked to a broad spectrum of opportunistic bacterial and fungal infections [1,2]. Inadequate insulin secretion or action causes an increase in blood glucose which, in turn, causes a series of metabolic and physiological abnormalities in the organs [3,4]. According to 2019 report, 1.5 million deaths were primarily associated with diabetes, and 48% of those deaths happened before the age of 70 [5]. The International Diabetes Federation (IDF) estimates that nearly 10% (537 million) of adults aged 20–79 had DM in 2021 [6]. Significant evidence supports the correlation between diabetes mellitus and an increased risk of infection. Many infections are more prevalent in diabetic individuals; some occur almost exclusively in this population. Other infections are more severe and are linked with an increased risk of complications in diabetic people. Since the immune systems of diabetic patients are changed in many ways, patients with diabetes are more prone to microbiological infections, such as pulmonary TB (Tuberculosis), urinary tract infections, pneumonia, and soft tissue and skin infections [7–10]. Patients who have diabetes with uncontrolled hyperglycaemia have weakened immune systems and are referred to as immune dysfunctional or sometimes immunocompromised [11–13]. Therefore, it is hypothesized that hyperglycaemia is the primary cause of impaired immunity in DM patients. Multiple mechanistic pathways are engaged in the impaired immune systems of diabetic individuals [14,15]. However, each

infectious agent (microbe) causes a distinct type of illness. After extensive review research (Figure 1) this article attempts to provide a critical overview of the current understanding of the processes behind the increased sensitivity of diabetes to infectious illnesses, as well to describe the principal infectious diseases associated with this metabolic condition and its management.

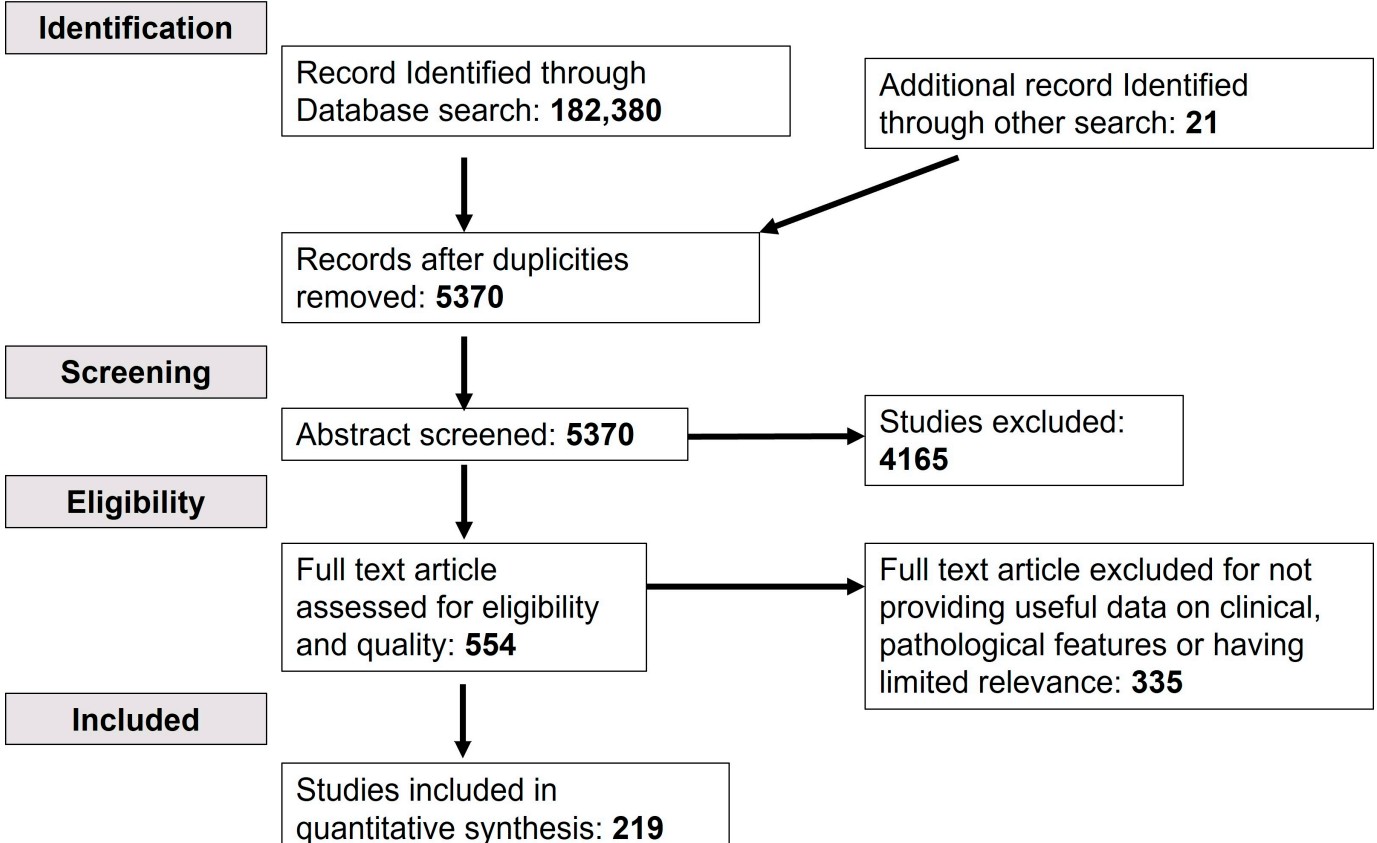

**Figure 1.** Prisma flow diagram: Preferred Reporting Items for Systematic Reviews (PRISMA) diagram for the selected 219 studies on a diabetes-associated infectious disease and its management.

## 2. Altered Immune Response in Diabetes

The human body has incredible defences against millions of bacteria, viruses, fungi, poisons, and parasites. The immune system normally protects against infections, but numerous disorders and deficiencies may impair it. Bacteria may infiltrate through open wounds and cause infections. Natural barriers, including intact skin and mucosal surfaces and the reactive oxygen species, cytokines, and chemokines within their mechanistic organization, help our defence system to fight against pathogens. Due to the immune system's inability to combat microorganisms, infections are a significant concern for people with diabetes. Numerous research have investigated diabetes-related pathways that decrease pathogen resistance [12]. These processes include the inhibition of cytokine production, abnormalities in phagocytosis, immune cell malfunction, and an inability to destroy microbials. The below section briefly outlines the immune dysfunctions associated with diabetes.

### 2.1. Oxidative Stress in Diabetes

In immune-suppressed diabetic conditions, increased glucose may raise ROS (Reactive Oxygen Species) and proinflammatory cytokines, limiting macrophage activity in systemic circulation [16–18]. Our recent observation documented GSH (glutathione) production and activated AGE (advanced glycation end products) and RAGE (receptor for advanced glycation end products) for the elevation of TGF-β (transforming growth factor- β) due

to excess hyperglycaemia [19,20]. If NADPH (nicotinamide adenine dinucleotide phosphate) is available, the enzyme GR (glutathione reductase) may recycle GSSH (oxidized glutathione). However, in diabetic individuals, NADPH is intensively used on the polyol pathway, resulting in a drop in GSH and an increase in ROS, which may change cytokine levels [17]. Diabetes mellites may also impair cellular immunity and weaken natural barriers. Studies have established that insulin insufficiency and hyperglycaemia are highly responsible for this phenomenon. Peripheral neuropathy, a consequence of DM, is related to the polyol pathway and sorbitol accumulation in the nerve tissue due to SDH (sorbitol dehydrogenase) deficiency [21,22]. Additionally, immune system abnormalities are also caused by excessive use of NADPH. The loss of NADPH results in reduced GSH, increased ROS, varying cytokine profiles and, eventually, advances immunological impairment [23].

## 2.2. Cytokine Response in Diabetes

In response to a pathogen, several immune cells, including macrophages and neutrophils, produce cytokines. The pattern recognition receptors of these cells identify bacterial cell wall components and release chemicals and nucleic acids to produce cytokines. These cells release cytokines that regulate and generate granulomas to confine and kill germs [16]. Granulomas include macrophages, multinucleated giant cells, CD4+ and CD8+ (clusters of differentiation) T-cells, B-cells, and neutrophils [24]. In immunocompetent people, bacterium and granulomatous immune system cells interact and release cytokines such as TNF-$\alpha$ (tumour necrosis factor alpha), IL-10 (interleukin), IL-6, IL-2, IL-12, and IFN-$\gamma$ (interferon-gamma) [25]. These cytokines play a crucial role in innate immunity. They are involved in diabetes pathogenesis and its affected, immune-mediated infection. In a study, the peripheral blood mononuclear cells (PBMCs) and isolated monocytes of patients with Type 1 diabetes (T1D) and Type 2 diabetes (T2D) released less IL-1$\beta$ than controls following lipopolysaccharides (LPS) stimulation [26,27]. Tessaro et al. found that the insulin treatment of bone-marrow-derived macrophages from diabetic mice boosted the production of TNF-$\alpha$ and IL-6 after LPS stimulation [28]. In another study, T1D monocytes from PBMCs produced less IL-1 and IL-6 than healthy donors. Interestingly, another study proved that PBMCs from non-diabetics are activated by anti-CD3 antibodies and subjected to high glucose levels, suppressing cytokines IL-2, IL-6, and IL-10 [29]. Furthermore, in another study, the authors observed that PBMCs taken from healthy people and stimulated with dextrose octreotide exhibited less IL-6 and IL-17A production, particularly in the CD14+ and CD16+ intermediate monocytes, suggesting multiple faulty immune responses in acute hyperglycaemia [30]. Compared to normal mice, obese leptin-receptor-deficient (DB/DB) mice and high-fat-diet-induced hyperglycaemic mice demonstrated decreased IL-22 levels [31]. Tan et al. found reduced IL-12 and IFN-$\gamma$ levels in diabetic PBMC cultures after infection with Burkholderia pseudomallei when compared with healthy donors. The PBMCs of diabetics had a more significant intracellular bacterial burden than healthy controls, indicating that hyperglycaemia affects the host's defense against infectious microorganisms. Recombinant IL-12 and IFN-$\gamma$ treatment decreased the bacterial load in diabetic PBMCs, demonstrating that the low production of IL-12 and IFN-$\gamma$ n diabetes inhibits the immune cells' ability to regulate bacterial development during an infection [32]. Therefore, hyperglycaemia in diabetes may reduce pathogen-fighting macrophage and leukocyte function.

As previously mentioned, hyperglycaemia also changes macrophage function. A study found that chronic hyperglycaemia impaired phagocytosis by affecting the activity of complement receptors on isolated monocytes. Long-term glucose sensitivity may impair the glycolic ability and reserves of macrophages. In contrast to hyperglycaemia, insulin deficiency in T2D has not been thoroughly studied regarding the function of macrophages against infections, necessitating research in this direction. [33]. In an in vitro study, elevated glucose inhibited antibacterial activity and phagocytosis in mouse bone marrow macrophages [34]. Additionally, peritoneal macrophages from diabetic animals showed decreased phagocytosis in the same research. Many studies have documented that, in diabetic

conditions, increased TNF-α expression induces a local inflammatory response through a cascade of cytokines. This increases vascular permeability, recruiting macrophages and neutrophils to the site of infection [35,36]. In another in vitro study, insulin intervention restored alveolar macrophage phagocytosis and cytokine release. This study suggests that exogenous insulin in diabetes may boost immune cell activity to fight pathogens by activating TNF-α and IL-6 [37]. In contrast, it is also possible that TNF- and IL-6 may cause insulin resistance. Therefore, exogenous insulin should be administered under strict control [38,39]. Furthermore, elevated plasma levels of TNF-α and IL-6 were shown to be associated with DM-induced LV dysfunction [40]. Liu et al. found that resident peritoneal macrophages (RPMs) from DB/DB mice had substantially lower phagocytosis and adherence [41].

### 2.3. Neutrophil Function and Other Immune Abnormalities in Diabetes

In addition to the generation of ROS, immunoglobulin-mediated opsonization, and phagocytosis, hyperglycaemia also impairs neutrophil degranulation and the formation of neutrophil extracellular traps (NETs). In one study, researchers found that individuals with induced hyperglycaemia who were exposed to bacterial wall components displayed lower neutrophil degranulation. [42]. Following previous results, Joshi et al. found that hyperglycaemia reduced neutrophil activity to create NETs, making infections more likely [43]. Natural killer (NK) cells regulate conditions in the innate cellular immune system, similar to other resistant components. The diabetes-related metabolic disorder affects NK cell activity, particularly the (natural killer group 2D) NKG2D/ligand axis. On the other hand, hyperglycaemia-mediated ER (endoplasmic reticulum) stress may contribute to diabetes-related NK cell dysfunction. Further research is needed to identify the dynamics of this interaction and how anti-diabetic medications affect ER stress protein expression in diabetes [44].

### 2.4. Immune Abnormalities in T1 Diabetes

T1D is a congenital autoimmune disorder produced by autoreactive CD4+ and CD8+ T-cells, which identify pancreatic antigens such as insulin or glutamic acid cecarboxylase (GAD) and kill insulin-producing β-cells [45]. How endogenous β-cell antigens become immunogenic is still open to exhaustive study. There is growing evidence that viral infection is a major environmental factor in the development and progression of T1D. Nevertheless, patients with T1D are also more susceptible to infections of the lower respiratory tract, urinary tract, and skin and mucosal membranes, as well as bacterial skin and mucous membrane infections and mycotic skin and mucous membrane infections [46]. Genome-wide association studies have helped achieve an understanding of genetic vulnerability. By contrast, immunological research has shown new immune-dysregulation mechanisms in the central tolerance, apoptotic pathways, or peripheral tolerance mediated by regulatory T-cells in T1D [47–49].

Thus far, several lines of evidence imply that diabetes-related comorbidities significantly contribute to immune disorders. Many studies explored diabetic immune responsive function to explain that the putative vulnerability to infections and hyperglycaemia was shown to decrease immune response system against antimicrobial activity in various trials. However, most immune response studies on diabetes-related infections show persistent immunological impairment. The above discussion makes understanding the issue challenging and explains the significant findings of different epidemiological studies. Therefore, diabetes is a complicated disease with several metabolic problems, making it challenging to trace the increased infection susceptibility through a single pathway or cell type. Additionally, multiple dysregulated physiological reactions affect organ failure and the 10% mortality rate. Furthermore, many diabetic survivors eventually die from recurring, nosocomial, and secondary infections, despite having a history of recovery. Recent studies demonstrate that even after clinical "recovery" from a deadly invasive infection, innate and adaptive immune responses remain altered, causing chronic inflammation, immunological suppression, and bacterial persistence. Moreover, diabetes lowers immune cell function

directly; thus, diabetic patients demonstrated a decreased bactericidal clearance, increased infectious comorbidities, and an increased mortality. Since diabetes-associated immune abnormalities increase mortality, immunological modulatory medication may enhance patient outcomes. Over the next two decades, diabetic infectious mortality is expected to climb significantly higher due to the increasing older and obese populations, worldwide adoption of a Western diet and lifestyle, and multidrug-resistant bacterial development and persistence. Understanding the underlying mechanism of diabetes-induced immune cell abnormalities that remain after an infection attack helps to identify possible targeted treatment strategies to boost innate and adaptive immune function, minimize infectious complications, and improve diabetic survival from fatal invasive infectious diseases.

## 3. Invasive Infection Susceptibility and Management in Diabetes

### 3.1. Invasive Aspergillosis Infection

Diabetic patients are more prone to microbial infection, particularly fungal infections [12,13,46]. A significant mortality rate is associated with invasive aspergillosis infection, a form of fatal fungal infection [50]. The mortality rate of invasive fungal illnesses is approximately 10 to 15%, with aspergillosis infection accounting for 42 to 64% of deaths in critically sick patients [51]. Invasive aspergillosis is caused by the mould *Aspergillus*, which causes different infections and allergy disorders depending on human immunity.

*Aspergillus fungi* are ubiquitous in outdoor and indoor environments. *Aspergillus* spores are inhaled regularly but do not cause any illness. In immune-dysfunctional diabetic individuals, *Aspergillus* causes lung infections, allergic responses, and organ infections [52]. *Aspergillus* spores cause moderate to severe lung infections. Aspergillosis becomes dangerous when it spreads to the blood vessels. Invasive pulmonary aspergillosis (IPA), which was first described in 1953, spreads rapidly under immunosuppression or chemotherapy. IPA is a type of aspergillosis in immunocompromised people, and it often infects critically sick patients comorbid with chronic obstructive pulmonary disease. Low white blood cell levels, lung cavities, asthma, cystic fibrosis, and long-term use of steroid medication may also induce IPA. *Aspergillus* causes fever, cough, asthma, shortness of breath, headaches, joint discomfort, and skin lesions. Aspergillosis may cause lung haemorrhage and extend to the heart, kidneys, and brain [53–55].

A cohort of 14 patients with CNS (central nervous system) aspergillosis who were admitted consecutively to Massachusetts General Hospital (MGH) from 2000 to 2011 was evaluated for the link between diabetes and CNS aspergillosis. Additionally, as per the literature, 123 more cases were also reviewed. They found that many individuals with CNS aspergillosis had little immune suppression; however, diabetes was linked to many paranasal instances. Primary paranasal aspergillosis patients had diabetes (17.9%). Based on the outcomes of surgical and non-surgical treatments of patients, it was assumed that there is an importance of surgical intervention in the management of CNS aspergillosis; however, there was also a possibility that the patients who underwent surgery were in better clinical condition. The good survival rates with neurosurgery reported in the literature might also be biased because it is more common to publish significant results. Until more efficacious treatment becomes available, the combination of effective antifungal therapy with neurosurgery should be strongly considered [56].

Another case study noted weight loss in 66% of *Aspergillus* individuals (15 out of 23). The amount of weight lost varied from 3.3 to 43 pounds (22 ± 3 lbs (mean ± SEM)). Eight out of twenty-three (34%) immunocompromised individuals with an invasive *Aspergillus* infection had diabetes ($p = 0.05$). Twenty-one out of twenty-three individuals reported weight loss, diabetes, or a low lymphocyte count. According to this study, weight loss and diabetes are risk factors for invasive aspergillosis and are associated with a high fatality rate [57].

One more case study reported that a 45 year old man with T2D was hospitalized with minor lumbar discomfort, irregular, low-grade fever, and intermittent dysuria for 3 months. This T2D patient developed unilateral renal aspergillosis after undergoing intracorporeal

pneumatic lithotripsy. The patient had minor lumbar discomfort and frequent urination of soft, white lumps. A microscopic analysis of the soft, white lumps, discharged with urine, indicated fungal hyphae and *Aspergillus fumigatus* infection. Initial treatment with amphotericin B was discontinued after two weeks as the patient was unable to take the medication as prescribed. His subsequent treatment with oral itraconazole was effective [58].

Another rare yet dangerous rhinosinusitis complication as reported with sinus–orbital apex aspergillosis. Clinical and radiological findings were insufficient to diagnose this disease. The case study was of a 64 year old woman with sinus–orbital apex syndrome and had a history of uncontrollably worsening diabetes. MRI (magnetic resonance imaging) indicated a right orbital apex mass encroaching on the right posterior ethmoid sinus, sphenoid sinus, and cavernous sinus. Functional endoscopic sinus surgery was conducted, and a sample of the lesion tissue indicated the presence of *Aspergillus*. One year after a surgical excision and antifungal and anticoagulant therapies, the patient was still observed to be quiescent without recurrence [59].

Another case study documented a 29 year old male with Type 1 diabetes who went to the ER after one week of chest discomfort and shortness of breath. The patient was taking pantoprazole and insulin. He also claimed to have been vaping marijuana every day for 18 months to ease diabetes-related neuropathic pain. DNA sequencing validated *Aspergillus rugolosa* in wedge-resection samples. Pleural fluid samples from that patient also developed *Aspergillus fumigatus*. Surgical excision and voriconazole medication healed the patient's infection. This period of therapy is consistent with successful treatments for chronic, necrotizing pulmonary aspergillosis and *Aspergillus* empyema. Incidentally, the experts also cited case reports of immunocompromised individuals with invasive Aspergillosis infections and marijuana use [60].

In another case study, a 54 year old woman with moderate, non-insulin-dependent diabetes mellites (NIDDM) and endometrial cancer was hospitalized for a routine laparoscopic cholecystectomy. She experienced duodenal perforation and peritonitis. Abdominal cultures indicated the presence of Candida albicans, C. glabrata, and C. tropicalis; the antibiotic regime was changed to piperacillin, tazobactam, and fluconazole. She required dialysis for prolonged hemodynamic instability and renal failure. Her blood, sputum, and abdomen were cultured, and ciprofloxacin was added. Amphotericin-B was added when *A. fumigatus* was identified. She had circulatory, respiratory, hepatic, and renal failure. Yet again, A. fumigatus was identified in the sputum cultures. No cancers or immunological diseases, including HIV, were detected. The antifungal treatment included the new antifungal drug, caspofungin. In spite of this, the patient developed resistant septic shock with progressive multiple organ dysfunction syndrome and died [61].

### 3.2. Invasive Zygomycosis

According to the literature, up to 44% of diabetic patients with zygomycosis die [62]; similarly, 25% of children with ketoacidosis and zygomycosis die [63] after developing the condition. On the other hand, more recent data from France revealed a case-fatality rate at hospital release of 9.2% in diabetic patients who remained relatively stable over a decade [64]. Zygomycosis is an invasive fungal infection occurring from zygomycetes, a division of zygomycota that includes terrestrial, parasitic, and symbiotic fungi. This category of fungus is a medically significant, opportunistic organism. This family of fungi produces aseptate or pauciseptate, broad, irregularly branched, ribbon-like hyphae and reproduces sexually through zygospores [65]. Zygomycosis is a life-threatening infection with a 30% mortality rate, particularly for infants and newborns, with various patterns of involvement at different ages [66]. Patients with diabetes, burns, trauma, and surgery, as well as those who take deferoxamine medication (for aluminium toxicity or hemochromatosis), are more prone to zygomycosis. Epidemiological studies have revealed higher instances of zygomycosis cases in developing countries [67]. Similar to other fungal infections, an impaired immune response is a significant cause of zygomycosis [62,68]. *Zygomycetes* may be ingested, inhaled, or inoculated. Zygomycosis causes pulmonary, rhino-orbital–cerebral,

cutaneous, and disseminated diseases in immunocompromised individuals. *Rhizopus oryzae*, *Rhizopus microsporus*, *Rhizopus homothallicus*, *Basidiobolus ranarum*, and *Conidiobolus incongruus* are common etiological agents [69–73].

According to Roden et al., among the 929 documented instances of zygomycosis, diabetic individuals accounted for 36% of the overall population, with Type 1 diabetes accounting for 20% of those cases. Moreover, 48% of Type 1 and 34% of people with T2D had ketoacidosis. The same authors later found a reduction in the number of diabetic individuals experiencing zygomycosis over time [62]. Researchers speculated that the extensive use of statins in the diabetic population, at least in the Western world, may have reduced the risk of zygomycosis [74]. Lovastatin has both in vitro and in vivo action against zygomycetes/zygomycosis, and statins are immunomodulators that may help combat against zygomycosis [75–77]. A recent report from a tertiary-care centre in North India highlighted diabetes-associated zygomycosis in the developing world. Out of 178 cases, 131 were in uncontrolled diabetics (73.6%); more notably, 56 cases (42.7% of diabetes cases) were caused by zygomycosis [67]. Undoubtedly, diabetes is another high risk factor for zygomycosis. Diabetes was revealed to be an independent risk factor for zygomycosis in a case study of 27 leukaemia and bone marrow transplant patients [78]. Uçkay et al. recently revealed that hyperglycaemia affects zygomycosis in solid organ transplant patients [79]. Additionally, zygomycosis cases are often associated with rhino-cerebral infection (35–50%). Rhino-cerebral zygomycosis is more common among diabetic individuals. The condition begins with sinus discomfort, soft tissue enlargement, and the necrosis of adjacent tissues due to thrombosis of the blood arteries. Moreover, associated vision loss or blurring indicates optic or ocular nerve impairment. The symptoms include nasal blockages, headaches, anosmia, necrotic facial lesions, double vision, and diminished awareness. Severe impairment of chemotaxis and neutrophil adhesion is seen, resulting in the incapacity of the neutrophils to combat the germination of fungal spores or eliminate growing fungal hyphae. Rhino-cerebral zygomycosis causes diabetic ketoacidosis. Patients with rhino-cerebral zygomycosis due to diabetic ketoacidosis have altered psychological conditions [80,81]. A cohort study of the 1994–2006 California investigation found 41 instances of rhino-orbital and rhino-orbital–cerebral zygomycosis. Approximately 59% of the patients had a documented history of diabetes mellitus, and approximately 21% of the 14 patients with corticosteroid-induced diabetes were taking diabetic medication [82]. Another study confirmed that, out of 179 rhino-cerebral zygomycosis patients, 60% had diabetes. Indian research, including on primarily diabetic individuals, found a 79.6% survival rate with amphotericin B (AmB) treatment and surgery [67]. Overall, the case-fatality rate of rhino-cerebral patients treated with antifungals alone drops from 70% to 14% [83].

Another unfavourable clinical case report documented a patient with persistent hypoglycaemia from diabetic breakdown, reported after TB therapy with minimal respiratory improvement. Considering the potential cellular malfunction of macrophages and neutrophils in diabetes, as well as the increased probability of invasive infection in immunosuppressed individuals, it may be speculated that the zygomycete caused the initial lung infection, which then spread to the CNS due to her diabetes-related compromised immune system. At admission, she showed severe tissue damage and necrosis, suggesting she had the condition one month earlier. Diagnostic issues, delayed surgery, and early antifungal medication directly affected this patient's deadly prognosis. Three days before death, histopathological and culture data were available. Therefore, amphotericin B treatment was initiated. This case highlighted that the lack of well-designed research describing the natural progression of zygomycosis: simultaneous lung, sinus, and CNS impairment, in addition to a link to pulmonary TB, had a contributory role in this patient's demise [84]. Additionally, a clinicopathological report by Challa et al. recorded that six out of the seven patients with pulmonary zygomycosis had diabetes [73], while statistical data suggest that unattended diabetes mellitus (in 73.6% of cases) is the most contributory risk factor for all kinds of zygomycosis (Odds Ratio 1.5–8.0), apart from renal zygomycosis [67].

### 3.3. Invasive Pneumococcal Infection

The invasive pneumococcal disease occurs when *Streptococcus pneumoniae* is isolated from a typically sterile location such as blood, cerebrospinal, joint, or pleural fluid. Infections caused by *Streptococcus pneumoniae* are a significant cause of morbidity and mortality. This Gram-positive bacterium causes life-threatening upper and lower respiratory infections, sepsis, and meningitis [85]. Invasive pneumococcal disease is more likely in those with certain medical conditions, poor socioeconomic status, or high-risk habits, including smoking and alcohol consumption [86,87]. Individuals with chronic illnesses have a two- to eightfold greater risk of dying from invasive pneumococcal disease when compared to those without chronic conditions, and older adults have a higher risk than younger folks [86].

Diabetes mellitus has been identified as an independent risk factor for developing respiratory tract infections [88]. It is a well-known risk factor for pneumococcal infection and predisposes people to nasopharyngeal colonization with the pneumococcus, which is linked to invasive infection [89]. Several molecular pathways have been proposed to explain the increased risk of pneumococcal infection in diabetics, the majority of which are mediated by the harmful consequences of hyperglycaemia. Saltzman et al. referred to these mechanisms as impaired humoral and cell immunity resulting from alterations in the polymorphonuclear leukocyte function, a diminished T-cell response, and diminished bactericidal serum activity [90,91]. Additional proposed pathways include pulmonary microangiopathy and other diabetes-related abnormalities in pulmonary tissue that compromise healthy lung function, contributing to decreased lung volume and restricted airflow [92–94]. Pneumococcal pneumonia is the most prevalent type of acute, bacterial, community-acquired pneumonia, affecting 8–50% of patients with pneumococcal infections. Of this number, 15–20% die despite antibiotic therapy [89]. Hence, pneumococcal immunization is recommended for people with diabetes regardless of whether they have been diagnosed with Type 1 or Type 2. Numerous in vitro investigations have demonstrated impaired immunity in diabetes, including reduced leukocyte activity [95]. Diabetic patients may have a decreased threshold for bacterial translocation into circulation due to vascular alterations or leukocyte malfunction. According to Moss et al., a less active inflammatory cascade may protect diabetic sepsis patients from acute respiratory distress syndrome [96]. Diabetes was linked to a reduced likelihood of acute respiratory distress syndrome in septic shock patients compared to non-diabetic individuals, according to their observations. A study proposed that the labelling and identification of comorbidities (such as heart disease) may have been more comprehensive for individuals with known diabetes due to more frequent hospitalizations. Regulation for comorbidity and other putative, confounding variables showed a very minimal effect on risk estimates, suggesting that none of the factors, including comorbidity, were significant confounders of the link between diabetes and pneumococcal bacteraemia outcome. It is unclear whether pneumococcal immunization may affect bacterial pneumonia disease. Intensive insulin treatment and statins have recently been related to the lower case fatality of bacteraemia [77,97]. Research conducted in North Jutland County, Denmark, reported that 46% of diabetic patients who received insulin during hospitalization had better survival rates [98]. In the US, the current pneumococcal vaccination contains 23 purified, capsular polysaccharide antigens covering 85–90% of the *Streptococcus pneumoniae* serotypes that cause invasive pneumococcal infections among children and adults. In addition to diabetes, secondary complications such as chronic cardiovascular, lung, and renal disorders increase risk in elderly patients. The ACIP (the Advisory Committee on Immunization Practices) recommends pneumococcal immunization for people with diabetes to minimize invasive disease [99]. The ACIP also specified that native American tribes with high rates of diabetes and invasive pneumococcal disease require more attention [100,101]. Notably, diabetic patients should not be revaccinated unless specific conditions occur. A one-time revaccination is advised for those >64. Otherwise, diabetes-related nephrotic syndrome, chronic renal illness, and

immunocompromised states, such as post-organ transplantation, can be considered causes for further repetitive immunization [102].

### 3.4. Invasive Rhinosinusitis

Rhinosinusitis, often known as sinusitis, is a symptomatic inflammation of the nasal cavity mucosa and paranasal sinuses that may be caused by allergens, viruses, bacteria, or fungi. Rhinosinusitis may be categorized depending on the timing of symptoms and the degree of inflammation: acute, subacute, or chronic [103]. Acute rhinosinusitis (ARS) is often characterized by the presence of symptoms for less than 12 weeks. It is further categorized depending on the length and assumed origin as follows: (a) viral rhinosinusitis (VRS) or (b) acute bacterial rhinosinusitis (ABRS) [104]. Four or more incidences of acute rhinosinusitis per year, with the improvement of symptoms between episodes, constitutes recurrent acute rhinosinusitis [105]. Subacute rhinosinusitis is characterized by symptoms that have lasted for longer than four weeks but less than three months [95]. Chronic rhinosinusitis (CRS) is characterized by the presence of symptoms for a minimum of three months and may occur with or without acute exacerbations and nasal polyps. Based on clinical phenotypes and endotype inflammatory patterns, CRS is further characterized as [104,106]:

(A). Phenotypes:

(a)     CRS without nasal polyps (CRSsNP)
(b)     CRS with nasal polyps (CRSwNP)

    (i)     Aspirin-exacerbated respiratory disease (AERD)
    (ii)     Allergic fungal rhinosinusitis (AFRS)

(B). Endotypes (predominant cytokines):

(a)     Type 1 (IFN-g)
(b)     Type 2 (IL-5, and IL-13)
(c)     Type 3 (IL-17)

Numerous risk factors for rhinosinusitis have been identified, including age, gender, smoking status, nasal polyps, allergic rhinitis, asthma, aspirin intolerance, infection, biofilms, gastroesophageal reflux disease (GERD), anatomical abnormalities of the upper airways, and the histological appearance of the sinus mucosa. Rhinosinusitis may affect people of any age group [105]. This study focuses on the prevalence and prevention of rhinosinusitis in the diabetic population. Most of the existing research on invasive fungal sinusitis consists of short case series and case reports. In evaluations of invasive fungal sinusitis, it was shown that between 47.8 and 60% of individuals had diabetes [107]. Due to the decreased innate and adaptive immunity associated with diabetes, as well as compromised wound healing and organ failure, comorbid diabetes in patients with CRS may lead to more severe symptoms and worse surgical outcomes in these individuals [108]. Diabetic individuals are more susceptible to various infections due to metabolic abnormalities [109]. In addition, new research has shown intestinal dysfunction to be one of the secondary complications associated with diabetes [110,111]. The skin and urinary tract are the most prevalent sites of infection in DM patients, including staphylococcal follicular skin infections, fungal infections, cellulitis, erysipelas, lower urinary tract infections, and acute pyelonephritis. DM infections are often multi-microbial in origin. Infections of the skin and soft tissues are usually caused by *S. aureus* in diabetic patient, although *P. aeruginosa* and other Gram-negative rods have been linked to worse outcomes in hospitalized patients. CRS patients with DM had an isolation rate of *P. aeruginosa* and other Gram-negative rods that was 3.5 times greater than CRS patients without DM. *S. aureus*, *S. pneumonia*, coagulase-negative staphylococcus, and anaerobe isolation rates were comparable in patients with and without DM. DM patients may be more likely to develop Gram-negative bacterial sinus infections and have considerably worse short-term postoperative quality of life [105].

A cohort study from AIMS hospital, India, documented hospitalized patients with invasive fungal rhinosinusitis and pre-existing or newly diagnosed diabetes between 2008

and 2015. They identified clinical traits, biochemical examinations, and treatment regimens utilizing hospital data. They found 22 diabetic individuals with invasive fungal sinusitis. Five patients presented with ketoacidosis, and all of them died before the completion of their hospital stay. Of these instances, 70% had a loss of vision in one eye. On day 30 of imaging, no patients demonstrated radiological improvement (including those who subsequently improved). Before two months of treatment, no radiological improvement was seen. The survival rate was 72.7% when the patients were discharged from the hospital, but it dropped to 57.8% six months after the completion of the study period. This clinical study highlighted that ketoacidosis could predict death in diabetic patients with invasive fungal sinusitis [112].

*3.5. Invasive Mucormycosis*

Mucormycosis affects immunologically impaired persons and causes substantial mortality and disability (>50%). Thrombosis and tissue necrosis are hallmarks of invasive mucormycosis. In most instances, the infection is progresses aggressively, resulting in mortality unless underlying risk factors (e.g., metabolic acidosis) are treated and vigorous antifungal therapy and surgical excision are instituted. This clinical study highlighted that ketoacidosis could predict death in diabetic patients with invasive fungal sinusitis [113]. Based on the clinical manifestation and anatomic location, invasive mucormycosis can be divided into six primary clinical types: (a) rhino-cerebral, (b) pulmonary, (c) cutaneous, (d) gastrointestinal, € disseminated, and (f) uncommon types, such kidney infection, endocarditis, osteomyelitis, and peritonitis. The sinuses (39%), lungs (24%), and skin (19%) are typical sites of invasive mucormycosis; in addition, approximately 23% of cases spread [114]. Country-by-country mucormycosis risk factors differ. Studies from France, the USA, and Greece show a shift from diabetes to haematological malignancies. According to the authors, 36% to 88% of mucormycosis sufferers are prone to diabetes mellitus, and the mortality rate for this condition is 52% [114,115]. Different nations have different mucormycosis epidemiology. In developed countries, the situation is infrequent and is usually encountered in individuals with diabetes mellitus, haematological malignancies receiving chemotherapy, and recipients of heterogeneous stem cell transplants. In underdeveloped countries, long-term and poorly controlled diabetes, diabetic ketoacidosis, or trauma are major risk factors for mucormycosis [114].

Skiada et al. showed in their case-control research that, in the most significant registry of the European Confederation of Medical Mycology Working Group, 230 cases related to diabetes mellitus patients were registered between 2005 and 2007; the statistics were around 9% [116]. During the same time, an extensive study from all French hospitals identified 101 instances of mucormycosis (60 proven/41 probable); these cases occurred predominantly in males (58%), and 23% suffered from diabetes. HIV (human immunodeficiency virus) infection does not cause mucormycosis unless it is accompanied by diabetes [117]. Maurer et al. found that diabetes and renal failure enhanced the likelihood of mucormycosis in solid organ transplant (SOT) patients [118]. Research from India found that 74% of mucormycosis patients had uncontrolled diabetes and, in 43% of these instances, diabetes was diagnosed for the first time. These data show the connection between poor diabetes management and socioeconomic level. Due to the high cost of healthcare for low-income people, they are often averted from the doctor until significant complications begin. Almost 80% of Type 2 diabetes fatalities occur in low- and middle-income groups. Many of these mucormycosis cases may have been prevented with proper diabetes management strategies [67,119].

Rhino-cerebral mucormycosis (ROCM) causes severe haemoptysis in diabetic individuals [62,116]. Patients with diabetic ketoacidosis or poorly managed diabetes mellitus are more likely to be infected with rhino-orbital–cerebral mucormycosis, and 67% of ROCM patients have diabetes mellitus. ROCM symptoms include eye and/or face inflammation, facial numbness, and blurry vision. Multiple cranial nerve palsies, unilateral periorbital face discomfort, orbital inflammation, eyelid oedema, blepharoptosis, proptosis, abrupt

ocular motility abnormalities, internal or external ophthalmoplegia, headache, and sudden vision loss are symptoms of ROCM [120]. According to a meta-analysis by Vaughan et al., DM was the most common underlying ailment in 175 ROCM cases published between 1994 and 2005 [121]. In a retrospective review from American teaching and tertiary care institutions, 83% of patients with rhino-orbital–cerebral mucormycosis (ROCM) had diabetes, while 41% had no history of diabetes. The prevalence of these cases was also significantly high during in COVID-19 [122].

Pulmonary mucormycosis (PM) is uncommon in diabetics but has substantial mortality and morbidity. A recent study demonstrated that Mucorales infections are prevalent in pulmonary cases and in 23.1% of diabetic patients [68]. Diabetes contributes to additional complications for individuals with comorbidities. In diabetes, pulmonary mucormycosis might show significant complications, including endobronchial or tracheal lesions. Endobronchial mucormycosis may induce lung collapse and blood vessel invasion, infiltration, amalgamation, nodules, cavitations, atelectasis, effusion, posterior tracheal band thickening, hilar or mediastinal lymphadenopathy, or unexceptional findings [123]. Rammaert et al. cited a series of 169 instances of healthcare-associated mucormycosis (HCM) identified from 1970 to 2008. Among those patients, solid organ transplantation (24%) (primarily due to graft-transmitted mucormycosis, affecting 60% of SOT recipients) and diabetes (22%) were the most common underlying diseases. In addition, 82–85% of individuals who developed endobronchial abrasions had diabetes. Cutaneous mucormycosis is the infection least likely to be connected with diabetes, though 26% of diabetic people have been diagnosed with it [124].

### 3.6. Herpes Zoster Infection

DM is a risk factor for both herpes zoster and post-herpetic neuralgia. In diabetes, post-herpetic neuralgia is more severe and chronic. This section addresses herpes zoster infections in DM and varicella-zoster virus (VZV) immunization in these patients. Diabetics have a higher risk of herpes zoster due to their lower cell-mediated immunity against VZV [125]. Though glycaemic management is often unrelated to cell-mediated immunity dysfunction, DM does not entirely impact VZV humoral immunity [125]. Interestingly, studies conducted worldwide revealed that T2D patients had a greater rate of herpes zoster infection incidence than age-matched controls [126–129]. T2D independently enhanced herpes zoster infection risk in a meta-analysis of 62 trials (relative risk 1.30, 95% confidence range 1.17–1.45) [130]. Herpes zoster patients sometimes have undetected T2D; hence, patients should be considered for testing in the presence of DM in particular [131]. T1D also may increase herpes zoster risk. Surprisingly, T1D may raise herpes zoster risk more than T2D [132]. On the other hand, a study documented that herpes zoster is more common in women with DM than in men [133]. Interestingly, older people with DM are more likely to have herpes zoster [134]. According to studies, thiazolidinediones, alpha-glucosidase inhibitors, DPP-4 (dipeptidyl peptidase-4) inhibitors, and insulin enhance the risk of herpes zoster. However, metformin and sulphonylureas are not likely to influence such probability [129]. DPP-4 inhibitors may suppress CD26, causing a susceptibility to herpes zoster infection. CD26 contains intrinsic DPP-4 activity and is found in T lymphocytes, B lymphocytes, and macrophages, activating and proliferating T cells [135]. Another study also confirmed that DPP-4 inhibitors might reduce cell-mediated immunity by inhibiting CD26 [136].

Patients with diabetes develop post-herpetic neuralgia more often [137]. T2D increased the likelihood of persistent post-zoster pain by 18%, documented by an extensive retrospective analysis of herpes zoster patients ($n$ = 420,515) [138]. Additionally, impaired glucose tolerance increases post-herpetic neuralgia risk [139]. Evidently, in a retrospective, population-based investigation, T2D patients with a herpes zoster infection used more healthcare resources than non-diabetics. T2D patients with herpes zoster had worsened glycaemic control, more outpatient visits, were prescribed more antivirals, were manifested with more hospitalization, and were unwell for longer [133]. On the other hand, regarding

herpes zoster infection, T2D patients have a lower quality of life than non-diabetics and heal more slowly [140].

## 4. Treatment Possibilities for Invasive Infection in Diabetes with Impaired Immunity

Most of the treatment strategies and techniques for aspergillosis, zygomycosis, mucormycosis, or herpes zoster do not change between diabetic and non-diabetic patients; however, the therapy does vary depending on the symptoms and tolerance level of the diabetic patient. Even if antifungal medicines and surgical treatment are available, invasive fungal infection in diabetic populations continues to be a disease with a high death rate. This is an unfortunate fact despite the availability of therapeutic options. The presence of ketoacidosis at the time of diagnosis is related to an increased likelihood of death. Imaging at many time points may be helpful in identifying intracranial expansion at an earlier phase. Following the diagnosis of the condition, focus should be on the clinical response and, subsequently, radiological imaging to direct the treatment strategy. If health care professionals have a better understanding of this atypical infectious condition, they may be able to lower fatality rates until more advanced treatment options become available. Here, we explored the many therapeutic options now available to fight against infectious ailments in diabetic patients.

### 4.1. Treatment Possibilities for Invasive Aspergillosis Infection

Treatment may involve monitoring, antifungal medicines, or surgery, depending on the aspergillosis infection. For improved infection treatment, the diagnostic technique should be accurate. Diagnosis of invasive pulmonary aspergillosis is not straightforward since harmful smear/sputum/respiratory secretions do not always indicate infection [141]. In the past, only a bronchoalveolar lavage and transbronchial biopsy could detect organisms in high-risk individuals utilizing contaminated smears [142]. The standard treatment for invasive pulmonary aspergillosis was Amphotericin B. However, studies demonstrated that Amphotericin B was ineffective due to its antifungal resistance and nephrotoxicity [143]. Previous research demonstrated that resistance to *A. fumigatus* varied from 0.6% to 29.6% [144]. Additionally, *A. flavus* and *A. ustus* isolates were resistant to amphotericin B [145–147]. Lipid formulations can increase their bioavailability; however, they are expensive, larger dosages do not always enhance effectiveness, lower levels might be hazardous, and high doses are not always tolerable [148].

Itraconazole is an antifungal drug used against Aspergillosis but that exhibits limitations in critically sick individuals with invasive pulmonary aspergillosis [149]. On the other hand, voriconazole showed more significant reactions and survival than amphotericin B [150]. Thus, voriconazole has been lately recommended and made accessible orally and intravenously. Oral dosing demonstrated a 90% absorption after 2 h [151]. Though voriconazole's low water solubility requires the use of sulfobutylether beta cyclodextrin sodium as a carrier, sulfobutylether beta cyclodextrin sodium causes moderate to severe renal impairment [152]. One animal trial revealed that sulfobutylether beta cyclodextrin sodium caused a low frequency of acute toxicity by inhibiting the function of the renal tubules and inducing liver necrosis. In severe individuals, oral voriconazole therapy might induce stomach reflux, gastro bleeding, intestine dysfunction, and poor drug absorption [153,154]. According to the research, long-term voriconazole therapy promotes yeast and mould resistance [155]. Posaconazole, another triazole, exhibited antifungal efficacy in pulmonary aspergillosis patients. Patients (both haematological and non-neutropenic) with refractory IPA who participated in a multicentric trial demonstrated a success rate of 42% when compared to controls [156]. Posaconazole only can be taken orally, making it less useful for critically sick patients with poor medication absorption. Echinocandins, micafungin, and anidulafungin are certified drugs for treating IPA, while caspofungin also demonstrated efficacy in IPA treatment. These medicines inhibit the production of fungal-cell-wall enzyme complex 1,3-D-glucan and outperform the antibiotic liposomal Amphotericin B in comparison [157,158]. Various clinical studies related to Aspergillosis

infection in diabetic patients and their treatment options, conducted in different countries, are summarised in Table 1. Following the discussion on the many treatments available to combat aspergillosis, it has become clear that a strategy for early clinical identification and selecting the correct treatment option for aspergillosis in diabetic patients need to be standardized.

**Table 1.** Recent studies on aspergillosis cases with diabetic patients.

| Condition | Causative Organism | Patient/Number of Patients | Symptoms | Country | Diagnosis | Treatment | References |
|---|---|---|---|---|---|---|---|
| Renal Aspergillosis | *Aspergillus fumigatus* | Forty-five-year-old diabetic male | Mild pain in the left lumbar region; irregular, low-grade fever; and occasional dysuria for 3 months. | Malaysia | Ultrasound scan and intravenous urogram (IVU) revealed a cystic lesion (4 × 3.9 cm) at the lower pole cortical region of the left kidney. | Amphotericin B for 2 weeks at 1.25 mg/kg/day, which was replaced by oral itraconazole (200 mg twice daily for two months and 100 mg twice daily for one month) as the person could not tolerate Amphotericin B. | [58] |
| Pulmonary aspergillosis | *Aspergillus* | (a) Forty-six-year-old diabetic female with hypertension | wheezing, chronic productive cough, and dyspnoea. | Iran | Elevated ESR [erythrocyte sedimentation rate] (126 mm/1st h) and fasting blood sugar: 229 mg/dl; the galactomannan level in serum was 1.7. Chest X-ray showed cavitary formation in the right upper lobe, spiral CT-scan of thorax with revealed large cavitation in anterior segment of right upper lobe, trans-bronchial lung biopsy. | Itraconazole | [159] |
| | | (b) Forty-five-year-old diabetic male smoker | Intermittent fever for a month, dyspnoea, dry cough, and weight loss. | | Galactomannan serum level was 1.8. Elevated ESR (66 mm/1st h), white blood cells:18,700/mm$^3$, Neut.: 88%, and blood sugar: 380 mg/dl. Cavitation in upper lobe nodules was revealed on chest X-ray and CT-scan of thorax. | Itraconazole | |
| Pulmonary aspergillosis | *Aspergillus versicolor* and *Aspergillus ochraceus* | Twenty-nine-year-old Type 1 diabetic male with marijuana use. | Chest pain for a week, weight loss, dyspnoea, fever, and night sweats for a year, | Canada | A radiograph of the chest showed pneumothorax and air space disease in the left lower lobe. Computed tomography of the chest showed consolidation and cavitation in the left lower lobe, annexing the pleura, pneumothorax, a chest tube and subcutaneous emphysema. | Surgery and six-months course of voriconazole. | [60] |

**Table 1.** *Cont.*

| Condition | Causative Organism | Patient/Number of Patients | Symptoms | Country | Diagnosis | Treatment | References |
|---|---|---|---|---|---|---|---|
| Pulmonary aspergillosis and mucormycosis | *Aspergillus fumigatus, Rhizopus arrhizus* | Seventy-nine-year-old diabetic, Latino male with hypertension and COVID-19 | Fever, rigors, dry cough, and dyspnoea for 10 days. | USA | Nasopharyngeal swab PCR (Polymerase Chain Reaction) test was positive for severe acute respiratory syndrome coronavirus-2 (SARS-CoV-2). Chest radiograph (CXR) revealed patchy bibasilar infiltrates. Computed tomography (CT) revealed moderate, bilateral, ground-glass opacities and infiltrates. On day 13, a broncho-alveolar lavage showed thick respiratory secretions. On day 19, the chest CT was repeated. | Five days of treatment with ceftriaxone, azithromycin, and remdesivir. A 7 day course of IV (intravenous) vancomycin 1250 mg every 8 h and IV ceftriaxone 1 g daily for treating ventilator-associated pneumonia. On day 14, 200 mg of IV voriconazole was administered twice daily for treating aspergillosis. From the 19th, day liposomal Amphotericin B was started for treating mucormycosis. On day 23, a tracheostomy was performed, and on day 25 was a percutaneous endoscopic gastrostomy. | [160] |
| Pulmonary aspergillosis | *Aspergillus fumigatus* | Forty-four-year-old diabetic male | Fever, haemoptysis, cough, and dyspnoea for 3 weeks. | India | Sputum analysis for fungus, chest X-ray, and contrast-enhanced computed tomography were performed, which showed multifocal areas of cavitation involving bilateral upper and lower lobes of the lung. Patient underwent flexible bronchoscopy and subsequent biopsy. | IV voriconazole, 6 mg/kg b.d., followed by 4 mg/kg b.d. for 2 weeks. | [161] |
| Pulmonary aspergillosis | *Aspergillus fumigatus* | Forty-five-year-old diabetic male with ketoacidosis and hypertension. | Flu-like illness with pyrexia. | UK | Chest radiography showed patchy consolidation and volume loss of the left lower zone. Bronchoscopy showed thick, white plaques over the left main bronchus, extending into left upper and lower lobe bronchi. | Initially started with intravenous cefotaxime, flucloxacillin, and metronidazole. From day, 5 piperacillin, gentamicin and metronidazole were administered. Intravenous fluconazole was given for candida albicans infection. From day 7, oral itraconazole, 200 ug b.d., and an intravenous amphotericin B colloidal dispersion (2 mg/kg bw increased to 4 mg/kg bw) were treated. Itraconazole was given for six months. | [162] |

**Table 1.** *Cont.*

| Condition | Causative Organism | Patient/Number of Patients | Symptoms | Country | Diagnosis | Treatment | References |
|---|---|---|---|---|---|---|---|
| Pulmonary aspergillosis | *Aspergillus* | Fifty-three-year-old diabetic female with keto-acidosis. | Fever, vomiting, and confusion. | Japan | A chest radiograph demonstrated infiltrates in the middle and lower zones of both lungs. A chest CT scan showed nodules with the halo sign and focal ground glass opacity and cavitation. Bronchoscopy showed thick, white plaques. | Intravenous ampicillin sodium/sulbactam and peramivir were initiated for her pneumonia. This was shifted to L-amphotericin B (100 mg/body per day for a month). | [163] |
| Pulmonary aspergillosis/ aspergilloma | *Aspergillus* sp. | Forty-five-year-old diabetic female | Chest pain, cough, decreased appetite, weight loss, and chronic haemoptysis. | Indonesia | X-ray and CT scan showed a mass in the upper lobe of left lung. Fine needle aspiration biopsy showed *Aspergillus* sp. surgery. | Initial treatment included the administration of codeine (3 × 10 mg), tranexamic acid (3 × 500 mg iv), novorapid (3 × 10 units), evemir (12 units, night), and Fluconazole (1 × 400 mg for 1 day, followed by 200 mg). Surgery was performed using pulmonary wedge resection | [164] |
| Rhino-oculo-cerebral aspergillus and mucor co-infection | Rhizopus & *Aspergillus flavus* | Forty-six-year-old diabetic male with hypertension and epistaxis in his past | proptosis of the left eye, ptosis, and diminution of vision for a month. Confused state | India | The radiological examination revealed opacification. Squamous cell carcinoma revealed on histopathology. X-ray and CT scan were performed for diagnosis. Sinoscopy confirmed a severely inflamed maxillary sinus. Fifteen days after the diagnosis of aspergillosis, histopathological studies revealed septate hyphae showing mucor infection. | Voriconazole treatment for seven days. After 15 days, Amphotericin B was given, but the patient expired due to intolerance. | [165] |

## 4.2. Treatment Possibilities for Invasive Zygomycosis

The early detection of zygomycosis using lesion histology and cultures improves the quality of care. Zygomycosis has nonspecific symptoms and may infect diverse organs. Surgical excision and antifungal medicines are used to treat cutaneous zygomycosis. Repeated surgical clearance or removal may be recommended when the infection is severe. Comparatively, cutaneous zygomycosis has a better prognosis; however, the death rate is 31%. [62]. Most zygomycosis survivors were addressed with a combination treatment of liposomal Amphotericin B and surgery. Posaconazole treatment is sometimes used as a last alternative and, in sporadic cases, hyperbaric oxygen therapy is also a consideration. A case study by Gargouri et al. (2019) showed a combination of standard Amphotericin B and caspofungin reversed clinical symptoms and, notably, ophthalmic involvement and facial paralysis, with a complete regression of homolateral ethmoid cell filling [166]. The study, which used molecular characterization of the biopsy samples, confirmed the presence of *Rhizopus arrhizus* filaments in the samples. In the same report, diabetes mellitus was the

predominant risk factor in 418 instances of zygomycosis in Mexico between 1982 and 2016. Out of 227 patients, 204 (89%) were treated with Amphotericin B, while 172 out of 191 (90%) also had surgery. The use of combination therapies improved the chances of survival for diabetic patients [115]. Different clinical studies in diabetic, zygomycotic patients and their treatment options, conducted in different countries, are briefed in Table 2. The preceding overview of existing treatments for zygomycosis emphasizes the need to create a strategy for the early clinical diagnosis and treatment of zygomycosis in diabetic patients.

**Table 2.** Recent studies on zygomycosis cases with diabetic patients and their treatment possibilities.

| Condition | Causative Organism | Patient/Number of Patients | Symptoms | Country | Diagnosis | Treatment | References |
|---|---|---|---|---|---|---|---|
| Gastrointestinal | Zygomycosis | Thirty-six-year-old diabetic female | Epigastric-pain, bilious vomiting, weight loss, fever, and constipation. | Iran | Endoscopy disclosed an extensive sub-mucosal haemorrhage. A biopsy demonstrated broad, aseptate fungal elements, a laparotomy showed rubbery, grey/brown necrotic tissue | Total gastrectomy with Roux-en-Y esophago-jejunostomy and surgical debridement, Amphotericin B lipid complex (1 mg/kg/dose). | [167] |
| Cutaneous | *Rhizopus arrhizus* | Twenty-six-year-old diabetic female with ketoacidosis. | Severe lower abdominal pain and intense thirst. | India | Biopsy specimen showed non-septate hyphae with right angle branching. | Local surgical debridement and Amphotericin B (4 mg/kg/day). | [168] |
| Cutaneous and rhino-orbitocerebral | *Apophysomyces elegans* | Fifty-year-old diabetic male | Facial pain and diplopia, right eye proptosis. | USA | Physical examination, CT, and MRI of the head showed right eye proptosis with inflammatory changes. Histopathology showed aseptate hyphae. | Liposomal Amphotericin B and multiple debridements. | [169] |
| Urinary bladder | Mucorales | Fifty-five-year-old diabetic male | Fever, dysuria, obstructive urinary symptoms, deranged creatinine, and flank pain. | India | Non-contrast CT scan showed right mild hydronephrosis. Cystoscopy showed yellowish-white material in urinary bladder. Histopathology studies revealed the fungal attack. | Posaconazole | [170] |
| Pulmonary | *Rhizopus oryzae* | Sixty-one-year-old diabetic male | Productive cough, fever, anorexia, and weakness. | India | A thick-walled cavity in the hilar region was seen from the chest X-ray. CECT (contrast-enhanced computed tomography) of the chest revealed a cavitating lesion. Histopathological studies showed fungal hyphae. | Amphotericin B for a cumulative dose of 3 g. | [171] |

### 4.3. Treatment Possibilities for Invasive Pneumococcal Infection

Traditionally, invasive pneumococcal disease (IPD) was treated with antibiotics. Despite the usage of antibiotics, mortality and morbidity related to IPD continue to be an issue due to the development of antibiotic resistance. Therefore, the prevention of pneumococcal infections should be given a high priority. Unlike the other conditions discussed here, the most promising prospect for preventing pneumococcal disease is vaccination.

Diabetic patients have a normal response to pneumococcal vaccination, which is a cost-effective preventive strategy [172,173]. Immunization can be performed using the pneumococcal polysaccharide vaccine (PPV), which can significantly reduce the morbidity and mortality caused by pneumococcal disease [174]. The pneumococcal polysaccharide vaccine includes 23 purified, capsular polysaccharide antigens representing 85–90% of the serotypes of *S. pneumoniae*. The 23 valent PPV (PPV23) can be given to the patient as a subcutaneous or intramuscular injection (preferably in the deltoid muscle or lateral mid-thigh).

The antibody response after a single dose of PPV begins 7–10 days after vaccination. IgM antibodies are the first to appear, but their presence can be measured only for a few months. IgG antibodies develop slowly, with a concentration peak even after 70–100 days, and are long-lasting, providing long-term immunity. The vaccine is generally safe, but mild, local side effects may be seen, with pain, soreness, erythema, warmth at the injection site, and fever among the most common side effects [89]. In 2017, Timothy et al. suggested that the pneumococcal polysaccharide vaccine should be given to patients with diabetes at diagnosis with up to two further five-yearly doses [175].

Pneumococcal disease is the leading cause of death in all age groups. According to an analysis of diseases, this can be prevented by vaccines. The high case fatality rate from bacteraemic pneumococcal disease demands effective preventive strategies, including immuno-prophylactic measures. Identifying populations at an increased risk for pneumococcal illness and/or its complications due to weakened immune systems, such as diabetes, should be a mandated factor in the design of vaccination programmes. Vaccination programmes rely primarily on a knowledge of the risk factors and a definition of subpopulations at an increased risk for IPD. Vaccination dose details for pneumococcal disease in different age groups during diabetic conditions are briefed in Table 3.

**Table 3.** Vaccination given to pneumococcal disease in diabetic patients.

| Age at First Dose (Months) | Primary Dose | Booster Dose | | Reference |
|---|---|---|---|---|
| | Pn7 | Pn7 | PPV23 | |
| 2–6 months | 3 doses (2/4/6 months) | 12–15 months of age | After 2 years of age. | [176–179] |
| 7–11 months | 2 doses (0/2 months) | 12–15 months of age | First dose: At least 6–8 weeks after the last dose of pnc7. | |
| 12–23 months | 2 doses (0/2 months) | None | Second dose: 5 years after the pn23 dose. | |
| ≥24 months | 2 doses (0/2 months) | None | - | [89,176–179] |
| 5–64 years | None | None | None | [89,180] |
| >64 years | None | None | Second dose if the vaccine was administered >5 years ago. | |

Pcnt-7,7-polyvalent conjugate vaccine (pneumococcal conjugate vaccine). PPV23-polyvalent polysaccharide vaccine (pneumococcal vaccine).

### 4.4. Treatment Possibilities for Invasive Rhinosinusitis

Rhinosinusitis infections can be treated with oral antibiotics, injectable medicines, and topical antibiotics (50). Theoretically, topical antibiotics provide the advantages of high, local drug concentrations with less systemic absorption, a lower cost, and reduced morbidity. Recent research on diabetic individuals in Taiwan demonstrated that therapy with a DPP-4 inhibitor (DPP4i) might reduce the incidence of CRS in diabetic patients. Since 2006, when DPP4i became accessible for treating T2D, meta-analysis studies have shown an elevated risk of all infections. DPP4i are all tiny, orally bioavailable compounds

that interact with the active site of DPP4 without interfering with other known activities, including its immune-system-related effects. In this research, DPP4i ($\geq$31 cDDD) reduced CRS in Asian DM patients. Higher doses proved advantageous for the metabolism and immune response of diabetic patients. In this trial research, higher Diabetes Complications Severity Index (DCSI) scores also demonstrated the beneficial effect of DPP4i usage, taking into account the change in inflammation as well as glucose management [181,182]. In the case of diabetic rhinosinusitis, further clinical research must be performed to identify potential emerging treatment agents with which to combat this devastating infection.

*4.5. Treatment Possibilities for Invasive Mucormycosis*

Mucormycosis is challenging to diagnose clinically, which complicates its epidemiology. Patients are frequently treated for mucormycosis without histology or microbiological confirmation. Modern, less invasive procedures for safer tissue sampling and genetic tools for pathogen detection may soon clarify many unclear clinical cases. Chamilos et al. reported that numerous lung nodules and pleural effusions upon first CT (computed tomography) scans were independent predictors of pulmonary mucormycosis [183,184]. Preoperative, contrast-enhanced CT scans help to characterize the severity and complications of the infection. CT scans demonstrate oedematous mucosa, fluid-filled, ethmoid sinuses, and deterioration of the periorbital tissue and bone border. Sinus CT is the ideal imaging modality for assessing invasion, although bone damage is commonly visible only after soft-tissue necrosis. MR imaging can detect the intradural and intracranial extent of infection, cavernous sinus thrombosis, and internal carotid artery thrombosis. A contrast-enhanced MRI may show perineural infection spread. MRI is more sensitive than CT scans for detecting orbital, soft-tissue infections. Early infected patients may have standard MR imaging, and high-risk patients should always have surgical exploration with biopsy analysis. Imaging investigations are nonspecific for detecting infectious species, which need histological confirmation of fungal tissue invasion. Given the fast progression of invasive mucormycosis and the increased mortality when the fungus reaches the skull, every diabetic patient with a headache and visual abnormalities should be evaluated with imaging and nasal endoscopy to rule out mucormycosis [114,185].

It is hypothesized that the restoration of inherent immunosuppression is also critical. Granulocyte colony-stimulating factor and GM-CSF (granulocyte-macrophage colony-stimulating factor and sargramostim) have been used to enhance neutrophil quantity and function. In specific case reports, interferon-g has also been employed since it stimulates the function of monocytes, macrophages, and neutrophils [186]. In diabetic individuals, good management of hyperglycaemia and ketoacidosis improved the efficacy of statins against certain mucormycosis, which may be supported by immunosuppressive therapy. Recent studies show that unbound iron in serum tends to lead diabetic individuals with ketoacidosis to mucormycosis. Patients with diabetic ketoacidosis (DKA) are susceptible to mucormycosis, supporting the significance of iron intake in the disease's aetiology. High quantities of free iron in the blood of DKA patient stimulated *R. oryzae* development at an acidic pH (7.3–6.88) but not at an alkaline pH (7.78–8.38) [187]. Adding exogenous iron to serum helped *R. oryzae* grow in acidic environments, but not at a pH of 7.4 [188]. According to the European Conference on Infections in Leukemia (ECIL) and the European Confederation of Medical Mycology (ECMM), Amphotericin B (L-AmB) and Amphotericin B lipid complex (ABLC) can be strongly recommended as first-line treatments for mucormycosis [189].

In addition, scientists have demonstrated that inhibiting the GRP78 (glucose-regulated protein) cell receptor using GRP78-specific immune serum may protect diabetic mice against mucormycosis [176]. Rezafungin, SCY-078 (Ibrexafungerp), orolofim, and encochleated Amphotericin B are a few of the novel antifungal medications undergoing clinical trials [177]. ISZ, a novel azole, was licensed to treat mucormycosis in the United States and Europe in 2015 [178]. Other antifungal medicines that are active against Mucorales are now in the research and development stage. VT-1161 is a new inhibitor of fungal CYP 51 (sterol 14$\alpha$-

demethylase) with anti-Mucorales action observed in in vitro. Using VT-1161 as a curative or preventative therapy prolonged the survival of the neutropenic mouse model of *Rhizopus arrhizus* [179]. The antifungal drug APX001A (Fosmanogepix) (previously E1210) targets the protein Gwt1, which is a glycosyl-phosphatidyl inositol post-translational modification pathway surface protein. Several authors demonstrated that APX001A is equally effective as AmB in protecting mice against *R. delamar* [190,191]. PC1244, a novel, long-acting fungicidal azole, has demonstrated antifungal efficacy against Mucorales, both in in vitro and in vivo [192]. Ibrahim et al. demonstrated that deferiprone protected diabetic mice against mucormycosis [193]. Due to a lack of evidence of its effectiveness, deferasirox is not presently indicated for first-line treatment. Nonetheless, it might be an effective means of enhancing the effectiveness of antifungal therapy [194]. In the end, for the diabetic mucormycosis patient, the ECMM and ECIL strongly recommend surgery and control of underlying disease, including the management of ketoacidosis and hyperglycaemia. They also highly recommended the modulation of corticosteroids and immune-boosting drugs and, if possible, the combinational treatment of an antimicrobial agent with surgery in diabetic patients. Different clinical studies and their treatment strategies, conducted in different countries on diabetic mucormycotic patients, are provided in Table 4.

**Table 4.** Recent studies on mucormycosis cases with diabetic patients.

| Underlying Diseases | Organ/Region Infected with Fungus | Objective | Patient/Number of Patients | Country | Methodology | Results and Conclusion | Reference |
|---|---|---|---|---|---|---|---|
| Diabetes | Lungs | Pulmonary mucormycosis and tuberculosis | A diabetic case with fungal co-infection. | A fifty-six-year-old female | Netherlands | X-ray, CT scan, RTPCR, and lobectomy. | Treatment with TB and mycosis medicines resulted in little side effects. Patients with diabetes should undergo testing for certain co-infections | [195] |
| Diabetes mellitus | Eye | Mucormycosis | Infarction of the optic nerve caused by mucormycosis in a diabetic patient. | A fifty-one-year-old male | USA | MRI, exenteration and sinus debridement. | Extensive infarction of the left optic nerve with inflammation of the ipsilateral and periorbital adnexa. Histopathology demonstrated the presence of mucormycosis. | [196] |
| Renal failure and diabetes mellitus | Eye | Mucormycosis | ROCM detected in an ocular nerve infection case. | A thirty-four-year-old man. | Taiwan | Ophthalmic and neurological examination, CSF (cerebrospinal fluid) examination, and MRI. | There were black eschars going from the bilateral canthi to the vascular area. It extended to cerebral and bilateral ophthalmic nerves. People with im-munocompromised patients might consider ROCM if they have neuro-ophthalmological symptoms. | [197] |

**Table 4.** *Cont.*

| Underlying Diseases | Organ/Region Infected with Fungus | Objective | Patient/Number of Patients | Country | Methodology | Results and Conclusion | Reference |
|---|---|---|---|---|---|---|---|
| Diabetic ketoacidosis with ophthalmoplegia | Nostril region | Mucormycosis | An instance of recovery from mucormycosis infection. | A twenty-two-year-old women | USA | CT scan, nasoendoscopy, and biopsy. | Surgical excision of the right eye, paranasal sinuses, maxilla, and palate, suboccipital craniectomy, and shunt for hydrocephalus, followed by an 18 month course of antifungal medication. The chance of infection was increased with several surgical procedures. | [198] |
| Diabetes, kidney failure, myelodysplastic syndrome, and acute leukaemia | Cerebral region | Mucormycosis | Retrospective study of 36 cases with Mucormycosis | Thirty-six cases | Mexico | Surgical debridement, CT scan, and MRI. | Systemic and rhino-cerebral mucormycosis. The report suggested medicinal and surgical treatment. | [199] |
| HIV infection and diabetes | Cerebral region | Mucormycosis | Mucomycorsis with vasculitis in a diabetic case. | A fifty-four-year-old woman | Brazil | CSF analysis, CT scan, histopathologic analysis, and angiography with HR-VWI (high-resolution vessel wall imaging). | Vasculitis accompanied with inflammation. More research is necessary to evaluate the accuracy of mucormycosis tests. | [200] |
| Diabetes mellitus | Cerebral region | Mucormycorsis | Progressive ophthalmoplegia and blindness in infection. | Eighteen-year-old woman | USA | Surgical debridements, MRI, lumbar puncture, funduscopic examination, and surgical debridement. | Observation of fungal hyphae in the ophthalmic artery and optic nerve perineurals in the absence of substantial optic nerve inflammation. Diabetic individuals with ophthalmoplegia and blindness should be evaluated for infection. | [201] |
| Diabetes mellitus (three patients) and chronic leukaemia (one patient) | Cerebral region | mucormycosis | Examining fungal infections in four patients with underlying illnesses. | Four cases | Turkey | CT scan Otorhinolaryngologic examination | There were neurological abnormalities detected. Two patients had passed away. Investigating mucormycosis in ophthalmoplegia and ensuring quick diagnosis should be considered. | [202] |

**Table 4.** *Cont.*

| Underlying Diseases | Organ/Region Infected with Fungus | Objective | Patient/Number of Patients | Country | Methodology | Results and Conclusion | Reference |
|---|---|---|---|---|---|---|---|
| Diabetes mellitus with Cushing's syndrome | Cerebral region | Mucormycosis | Cushing's syndrome and solid tumours are associated with infection. | Forty-two-year-old woman | Mexico | CT scan Autopsy | Infarction of the left temporal lobe. The cause of the patient's death was determined to be a multihormonal pituitary adenoma with expansion to the sphenoid bone and sellar erosion. ACTH (ectopic adrenocorticotropic hormone) was detected in the left lung. The research established a correlation between ACTH and ectopic pulmonary tumours, pituitary apoplexy, and mucormycosis. | [203] |
| Diabetes mellitus and immunosuppression conditions | Cerebral region | Mucormycosis | Reginal differences in the infection and its causes. | - | Middle East and North Africa | Data collection | A total of 310 instances of infection. Most cases were associated with diabetes and immunosuppression. It is necessary to put into practice efficient treatment and preventive measures. | [204] |
| Diabetes | Cerebral region | Mucormycosis | A case of diabetes infected with mucormycosis. | Elder man | Canada | CT scan Autopsy | Thrombosis with infection in cerebral region. Early diagnosis is the key to effective therapy. | [205] |
| Diabetes mellitus with Garcin syndrome | Cerebral region | Mucormycosis | Analysis of infection and tuberculosis meningitis in a case with underlying disease. | - | China | CT scan with X-ray | Tuberculosis meningitis developed to mucormycosis. Diagnostics should be first in identifying the infection. | [206] |
| Diabetes mellitus | Cerebral region | Mucormycosis | To determine the prevalence and risk factors of mucormycosis in individuals with diabetes mellitus. | Total of 162 patients | Iran | Detailed history, otorhinolaryngologic, ophthalmic, and neurologic examinations | A total of 30 individuals had diabetes (19 were women and 11 were men). Diabetes might be a risk factor for fungal infections. | [207] |

**Table 4.** *Cont.*

| Underlying Diseases | Organ/Region Infected with Fungus | Objective | Patient/Number of Patients | Country | Methodology | Results and Conclusion | Reference |
|---|---|---|---|---|---|---|---|
| Diabetes mellitus | Cerebral region | Mucormycosis | Identification of infection in diabetes patient with complication to acute infarction | Fifty-seven-year-old man | Iran | CT scan Biopsy | Subarachnoid haemorrhage associated with a stroke. The biopsy revealed a mucormycosis infection. Early actions are required to prevent severe consequences. | [208] |
| Diabetes mellitus | Sinus region | Rhizopusaarhisus | To estimate the distribution of infection and its associated factors. | A total of 208 cases | Iran | Sequencing and data collection. | From 2008 to 2014, there was an increase in infections. It is crucial to monitor and identify this infection. | [209] |
| Diabetes and non-diabetic patients | Rhino-orbito-cere-bral | Mucorales | To compare fungal infections in people with and without diabetes. | Total of 63 patients | Iran | Ophthalmic investigation, imaging studies, and biopsy. | Survival was recorded in 51% of diabetic patients and 70% of non-diabetic patients. Neither group's rate of vision survival differed from the other. | [210] |
| Diabetes mellitus, Malignancy, transplant | Rhino-orbital | Rhizopus | Mucormycosis was the subject of a prospective observational research that was carried out across 12 locations in India. | Total of 465 patients | India | Questionnaire analysis | Symptoms with a shorter duration. The shorter duration of antifungal medication and the use of Amphotericin B were independent risk factors for death. Diabetes was the primary risk factor. | [211] |

Table content taken [212].

### 4.6. Treatment Possibilities for Herpes Zoster

A live, attenuated vaccine (Zostavax) for controlling herpes zoster infection that targeted the Oka VZV strain was first approved in 2006 [213]. This vaccination can be considered for both diabetic and non-diabetic patients. Pilot research was performed at Kitano Hospital, a general hospital in Osaka, Japan, from May to November 2010, to assess the varicella-zoster virus-specific, cell-mediated immunity and humoral immuno-genicity against the live varicella vaccine (Oka strain). After immunization, varicella skin tests, immune-adherence hemagglutination tests, and interferon-gamma, enzyme-linked immune-spot assays were performed. Vaccine safety was examined using a 42 day survey and tracking zoster development over one year. In persons with or without diabetes, the live herpes zoster vaccination increased virus-specific, cell-mediated, and humoral protection. No systemic adverse response or herpes zoster development case was identified in any research subject [214].

*4.7. The Role of Beneficial Bacteria in Diabetic Immuno-Comptonization*

The microbiome plays a crucial role in the development of the immune system and its maintenance. However, dysbiosis of the microbiome is observed in the diabetic condition. This leads to the modulation of local and systemic immunity. However, discrepancies exist among reported studies involving microbes [215]. Fortunately, despite the extensive discussion of bacterial infections in diabetes in the preceding section, some beneficial bacteria play a positive function in our system. Genera of Bifidobacterium and Bacteroides, Akkermansia, Faecalibacterium, and Roseburia are found to be beneficial even in T2D conditions. In contrast, the genera of Fusobacterium, Blautia, and Ruminococcus were reported to be negatively associated with T2D. On the contrary, the genus Lactobacillus, which encompasses a highly diversified species, shows both positive and negative associations with hyperglycaemic conditions. Beneficial gut microbiota competitively inhibit pathogen growth by forming siderophores, producing antimicrobial enzymes/metabolites, and competing for nutrition [216]. Henceforth, probiotics have been used in combinatorial therapy as adjunctive or in prophylactic treatments of fungal diseases. They can also be a treatment approach for infectious diseases associated with diabetes [217].

## 5. Conclusions and Future Direction

The chance of developing certain invasive infections is well-known in the diabetic condition. When dealing with diabetic patients with facial symptoms, medical professionals should proceed with extreme care. Purulent nasal discharge may indicate an active fungal infection, although diagnosis is sometimes challenging. Ketoacidosis undoubtedly increases the risk of fatal infections, yet most infections occur in poorly balanced diabetics without ketoacidosis. In diabetic patients, diagnostic delay is the leading cause of morbidity and fatality. Unfortunately, there are currently no vaccinations approved for use against infectious diseases, and the number of medications that may be used to treat potentially fatal infections is very restricted. Despite the availability of potentially beneficial therapeutic options, additional treatments have not been pursued in recent years. It is imperative that more research is conducted to achieve a more in-depth understanding of the pathogenic processes and the association between microbial organisms and the microenvironment that can be found in the human body. This understanding, in conjunction with the use of cutting-edge technology, such as CRISPR/Cas9 for the genetic alteration of the intricate genome of the microbes, will undoubtedly lead to the creation of improved antimicrobial drugs, alternative treatment options, and vaccines.

**Author Contributions:** Conceptualization, S.B. and K.S.; writing—original draft preparation, A.K., G.H., J.N., S.R.P. and S.B.; writing—review and editing, S.R.P. and S.B.; visualization, A.K., G.H., J.N., S.R.P., S.B. and K.S.; supervision, S.B. and K.S.; project administration, K.S.; resources, S.B. All authors have read and agreed to the published version of the manuscript.

**Funding:** This research did not receive any specific grant from funding agencies in the public, commercial, or not-for-profit sectors.

**Institutional Review Board Statement:** Not applicable.

**Informed Consent Statement:** Not applicable.

**Data Availability Statement:** Not applicable.

**Conflicts of Interest:** The authors declare no conflict of interest.

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
