# Peer review of "Management of Invasive Infections in Diabetes Mellitus: A Comprehensive Review"

_biologics, doi:10.3390/biologics3010004_

Round 1

Reviewer 1 Report

Khanam et al studied the management of infections in diabetic patients that are immunocompromised, to prone to accumulate infections. They discussed comprehensively the effect and treatment options for several infecting bacteria/fungi. However, the review should be improved for publications with these following suggestions.

1.       A standard PRISMA chart could be included by introducing  materials and methods.

2.       They should write clearly a paragraph initially with a heading how/possible mechanism of the excess glucose in diabetes patients contribute to immunocompromisation. Then they should discuss effect of infection in immune compromised conditions. The two events seems mixed up.

3.       NET, GAD,NIDDM, AIMS should be expanded. Before expansion IA came in the text (line 220)

4.       In line 127-128, TNFα and IL6 are proinflammatory molecules that induces inflammation. In various health conditions, scientists attempted to suppress these two cytokines to reduce inflammations.  If exogenous insulin activates these two cytokines, other immune mediated health abnormalities is expected. Is there any such report for external insulin use? If yes, they should be mentioned and discussed here.

5.       Line 132-134, sentence does not have a sense. Please rewrite this sentence.

6.       Line 141-144, please rewrite what authors actually wants to convey the message.

7.       Line 575, amphotericine should be written properly.

8.       The role of beneficial bacteria should be included in a separate section that counteracts diabetic  immunocompromisation and these infectious agents.

9.       Extensive English editing is required.

Author Response

Reviewer:1

The authors profusely thank the reviewers for complements and further suggestions to enhance the quality of this manuscript. The manuscript is now suitably revised, and the changes are incorporated and marked up using the “Track Changes”. The detailed responses to the reviewer’s comments are as follows:

  1. A standard PRISMA chart could be included by introducing materials and methods.

Response: As per the suggestion, A standard PRISMA chart could be included.

  1. They should write clearly a paragraph initially with a heading how/possible mechanism of the excess glucose in diabetes patients contribute to immune compromission. Then they should discuss effect of infection in immune compromised conditions. The two events seems mixed up.

Response: Now the section “immune response in diabetes” has been rewritten accordingly with subheadings (Page No: 3-6)

  1. NET, GAD, NIDDM, AIMS should be expanded. Before expansion IA came in the text (line 220)

Response: The abbreviations have been elaborated properly (Line No: 105, 111, 183).

  1. In line 127-128, TNFα and IL6 are proinflammatory molecules that induces inflammation. In various health conditions, scientists attempted to suppress these two cytokines to reduce inflammations. If exogenous insulin activates these two cytokines, other immune mediated health abnormalities is expected. Is there any such report for external insulin use? If yes, they should be mentioned and discussed here.

Response: As suggested, the health abnormalities upon TNFα and IL6 activation by exogenous insulin administration has been discussed with the citations in line no 97-99.

  1. Line 132-134, sentence does not have a sense. Please rewrite this sentence.

Response: The sentence has been rewritten.

  1. Line 141-144, please rewrite what authors actually wants to convey the message.

Response: The sentence has been rewritten.

  1. Line 575, amphotericine should be written properly.

Response: The word “amphotericin b” has been written correctly.

  1. The role of beneficial bacteria should be included in a separate section that counteracts diabetic immunocompromisation and these infectious agents.

Response: The role of beneficial bacteria should be included in a separate paragraph in the treatment section (Line No: 529-541)

  1. Extensive English editing is required.

Response: As suggested, we have extensively revised grammar and readability throughout the manuscript.

Reviewer 2 Report

The manuscript entitled “Management of invasive infections in immunocompromised diabetic populations: A comprehensive review” is a narrative review encompassing bacterial and fungus infection in a diabetes mellitus setting. The review reports on a relevant topic. However, some points deserve further attention before we proceed.

Major comments

 1)      The main idea of the manuscript needs to be reviewed, as follow: a) although diabetes mellitus is associated with immune system dysfunction from both innate and acquired immune system, the title including “immunocompromised diabetes mellitus” gives a sense of patients receiving immunosuppressive drugs, such as kidney transplant recipients. And this is not the case. b) the review covers mainly invasive fungal and bacterial infections (e.g., aspergillosis, zygomycosis, pneumococcal, rhinosinusitis, and mucormycosis) and their treatments. Adjust the title accordingly, as viral infections, such as herpes infections, are equally prevalent in diabetic patients and are not covered here. c) include a section on viral infection of diabetes mellitus.  

2)      Rather than quoting too many details throughout the manuscript (for example, on pages 4-5, lines 187-228), provide information on the odds ratio and whether there was a significant difference between diabetic and non-diabetic individuals when clinical manifestations and outcomes were compared.

3)      Likewise, how are treatments for aspergillosis, zygomycosis, pneumococcal, rhinosinusitis, and mucormycosis different between diabetics and non-diabetics?

4)      What is the horizon for the diagnosis, outcomes, prevention, and treatment of infections in diabetics to mitigate adverse mortality rates?           

5)      There is room to improve the manuscript from an English grammar point of view.            

Minor comments

1)      On Page 1, lines 37 & 39, review the data: “In 2019, up to 48% of patients aged 70 or younger were diagnosed with diabetes, and there were around 1.5 million DM-related fatalities”. Is the information about Forty-eight percent correct?

2)      On page 3, line 113, review the term “diabetic hyperglycemia”.

3)      On page 5, line 205, the words “3 cases” are repeated.

4)      On page 5, lines 213-215, review the sentences, as the numbers are discordant: “50 (67.5%) of 74 non-neurosurgery patients died. 54 of the 74 non-neurosurgery patients who got antifungal therapy died in hospital.”   

5)      Review the manuscript accordingly and verify that type of information.    

Author Response

Sub: Resubmission of revised review article

            Manuscript ID: biologics-2154906

   Title:Management of invasive infections in diabetic populations with immune dysfunction: A comprehensive review

    We are resubmitting our revised review article entitled “Management of invasive infections in diabetic populations with immune dysfunction: A comprehensive review” for consideration by Biologics.

   We were pleased by the overall positive review and general enthusiasm for our study. We confirm that this review work is original and has not been published elsewhere, nor is it currently under consideration for publication elsewhere.

We have thoroughly revised the manuscript to address the reviewers’ concerns and incorporate their suggestions. We hope that the revised manuscript is acceptable for publication. Listed below are our detailed responses to each criticism or suggestion, which are marked up using the “Track Changes” in the revised manuscript. 

     We confirm that neither this manuscript nor any parts of its content are currently under consideration or published in another journal. All authors have approved this manuscript and agree with its resubmission to “Biologics”.

  Thank you for giving us the opportunity to resubmission. I am pleased to resubmit the review manuscript herewith. I humbly request its consideration for publication in Biologics.

Reviewer:2

The authors profusely thank the reviewers for complements and further suggestions to enhance the quality of this manuscript. The manuscript is now suitably revised, and the changes are incorporated and marked up using the “Track Changes.” The detailed responses to the reviewer’s comments are as follows:

Major comments

  1. The main idea of the manuscript needs to be reviewed, as follow: a) although diabetes mellitus is associated with immune system dysfunction from both innate and acquired immune system, the title including “immunocompromised diabetes mellitus” gives a sense of patients receiving immunosuppressive drugs, such as kidney transplant recipients. And this is not the case. b) the review covers mainly invasive fungal and bacterial infections (e.g.,aspergillosis, zygomycosis, pneumococcal, rhinosinusitis, andmucormycosis) and their treatments. Adjust the title accordingly, as viral infections, such as herpes infections, are equally prevalent in diabetic patients and are not covered here. c) include a section on viral infection of diabetes mellitus.

Response: a) As per suggestion, the title of the manuscript has been reframed. “Management of invasive infections in diabetic populations with immune dysfunction: A comprehensive review.”

b), c) A section on herpes infections and viral infection in diabetic patients has been included (Line No: 365-387; 518-527).

  1. Rather than quoting too many details throughout the manuscript (for example, on pages 4-5, lines 187-228), provide information on the odds ratio and whether there was a significant difference between diabetic and non-diabetic individuals when clinical manifestations and outcomes were compared.

Response: Now, the portion has been concise with the significant findings from the study (Line No: 152-163).

  1. Likewise, how are treatments for aspergillosis, zygomycosis, pneumococcal, rhinosinusitis, and mucormycosis different between diabetics and non-diabetics?

Response: Treatment for Aspergillosis, zygomycosis, and mucormycosis does not differ between the diabetic or non-diabetic, but with respect to the symptoms and tolerance level by the affected diabetic individual, the treatment varies, which is mentioned in the manuscript (table 1).  In case of invasive pneumococcal infection, vaccination is mandatory to decrease the effect of mortality and morbidity in affected individuals. Since diabetic individuals have compromised immune systems. An effective vaccination program depends on the risk of the disease. Rhinosinusitis infections may be treated with oral antibiotics, injectable medicines, and topical antibiotics. According to Rammaert et al., 2012  and the American diabetes association.2019, CRS in diabetic patients have impaired innate and adaptive immunity along with compromised healing and organ failure compared to nondiabetic individuals. Hence studies have shown that therapy with Dipeptidyl peptidase-4 inhibitor (DPP4i) might reduce the incidence of CRS in diabetic patients.

We also mentioned that no treatment changes between the diabetic or non-diabetic patient in the treatment section (Line No: 390-398)

  1. What is the horizon for the diagnosis, outcomes, prevention, and treatment of infections in diabetics to mitigate adverse mortality rates?

Response: A section indicating “the horizon for the diagnosis, outcomes, prevention, and treatment of infections in diabetics to mitigate adverse mortality rates” has been mentioned in the conclusion part (Line No: 544-554).

  1. There is room to improve the manuscript from an English grammar point of view.

Response: As suggested, we have revised grammar and readability throughout the manuscript.

Minor comments

  1. On Page 1, lines 37 & 39, review the data: “In 2019, up to 48% of patients aged 70 or younger were diagnosed with diabetes, and there were around 1.5 million DM-related fatalities”. Is the information about Forty-eight percent correct?

Response: The sentence has been rewritten with a clear statement (Line No: 33-34).

  1. On page 3, line 113, review the term “diabetic hyperglycemia”.

Response: The term has been corrected.

  1. On page 5, line 205, the words “3 cases” are repeated.

Response: The repetitive word has been deleted.

  1. On page 5, lines 213-215, review the sentences, as the numbers are discordant: “50 (67.5%) of 74 non-neurosurgery patients died. 54 of the 74 non-neurosurgery patients who got antifungal therapy died in hospital.”

Response: The portion has been rewritten.

  1. Review the manuscript accordingly and verify that type of information.

Response: We are glad to have the reviewer’s overall positive review for the improvement of the review article.

Round 2

Reviewer 1 Report

                Now the paper is improved. But still there are some concerns.

1.       Many terms (AGE, RAGE, GSH, GSR, NET, etc)  in the text should be expanded or an abbreviation list should be included.

2.       Line 92,  “Accordingly, Spindler et al. 92 found that high blood glucose levelss inhibited immunological responses in healthy persons PBMCs generated with dextrose 93 octreotide, specifically in CD14+ and CD16+ intermediate monocytes [30]” should be rewritten”.

How high blood glucose level inhibited …..in healthy persons PBMC “…

3.        Line 97-98, “Recombinant… “ is against the message of the paper. If hyperglycemia induced IL2 and IFNß secretion  reduce bacterial development.  That suggests hyperglycemia inhibits infection!!

4.       In Line 140,  “insulin  …… even enhanced the inflammatory response, however through hyperglycemia??”” How it can happen or rewrite the sentence!!

5.       Line 144, “why the different epidemiological studies had have inconsistent findings”-author should explain the inconsistent findings” in this context.

6.       English grammars/editing, corrections are still necessary.

7.       Author should accept the mistakes in word files and then make the pdf file. The modified portions should be marked in highlighter that would be easier to reviewer to view it. No need to show which portions are deleted!!

Author Response

Reviewer:1

The authors profusely thank the reviewers for complements and further suggestions to enhance the quality of this manuscript. The manuscript is now suitably revised, and the changes are incorporated and marked up using the “red font color”. The detailed responses to the reviewer’s comments are as follows:

Comments:

  1. Many terms (AGE, RAGE, GSH, GSR, NET, etc)

in the text should be expanded, or an abbreviation list should be included.

Response: As per the reviewer’s suggestion, all abbreviations have been expanded in the text, and the corrections are highlighted with red font color.

  1. Line 92, “Accordingly, Spindler et al. 92 found that high blood glucose levels inhibited immunological responses in healthy persons PBMCs generated with dextrose 93 octreotide, specifically in CD14+ and CD16+ intermediate monocytes [30]” should be rewritten”.

How high blood glucose level inhibited …..in healthy persons PBMC “….

Response: The portion has been rewritten (Line: 106-109).

  1. Line 97-98, “Recombinant… “is against the message of the paper. If hyperglycemia induced IL2 and IFN- ß secretion reduce bacterial development.

That suggests hyperglycemia inhibits infection!!

Response: The sentence has been rewritten (Line: 115-118).

  1. In Line 140, “insulin …… even enhanced the inflammatory response, however through hyperglycemia??”” How it can happen or rewrite the sentence!!

Response: The portion has been rewritten (Line: 165-168 ).

  1. Line 144, “why the different epidemiological studies had have inconsistent findings”-author should explain the inconsistent findings” in this context.

Response: The statement has been rewritten (Line: 169-172).

  1. English grammars/editing, corrections are still necessary.

Response: As suggested, we have extensively revised grammar and readability throughout the manuscript with due diligence.

  1. Author should accept the mistakes in word files and then make the pdf file. The modified portions should be marked in highlighter that would be easier to reviewer to view it. No need to show which portions are deleted!

Response: As suggested, we have now highlighted the corrected part with red font color after accepting the track change. Though we have not made the color change on grammatical corrections since there are several little adjustments.

Reviewer 2 Report

The authors presented an improved version of the manuscript. However, there is still room to improve the manuscript from an English grammar point of view.    

Minor points:

1)     I suggest keeping the title more objective: “Management of invasive infections in diabetes mellitus: a narrative review. Likewise, avoid the terms “diabetes with immune dysfunction”, as these patients are not taking immunosuppressive drugs. 

2)     In Table 2, remove the city Minnesota, as other names from other countries are not described. Likewise, correct and standardize the names of the countries (for example, Texas, Netherland, etc) in Table 4.

3)     Include a list of abbreviations at the end of each table.

Author Response

Reviewer:2

The authors profusely thank the reviewers for complements and further suggestions to enhance the quality of this manuscript. The manuscript is now suitably revised, and the changes are incorporated and marked up using the “red font color”. The detailed responses to the reviewer’s comments are as follows:

Minor comments

  1. suggest keeping the title more objective: “Management of invasive infections in diabetes mellitus: a narrative review. Likewise, avoid the terms “diabetes with immune dysfunction”, as these patients are not taking immunosuppressive drugs.

Response: As per suggestion, the title of the manuscript has been reframed. “Management of invasive infections in Diabetes Mellitus: A comprehensive review.”

  1. In Table 2, remove the city Minnesota, as other names from other countries are not described. Likewise, correct and standardize the names of the countries (for example, Texas, Netherland, etc.) in Table 4.

Response: Now, in all the tables, we have used standardized places' names and highlighted them with red font color (different countries).

  1. Include a list of abbreviations at the end of each table.

Response: As per the reviewer’s suggestion all abbreviations have been expanded in the text and in the tables as well as the corrections are highlighted with red font color.